# Concept Complement Bottleneck Model for Interpretable Medical Image Diagnosis

**Hongmei Wang**                  HWANGFY@CONNECT.UST.HK
**Junlin Hou**                     CSEJLHOU@UST.HK
**Sunan He**                    SHEBD@CONNECT.UST.HK
**Shu Yang**                   SYANGCW@CONNECT.UST.HK
*Department of Computer Science and Engineering, Hong Kong University of Science and Technology, Hong Kong, China.*

**Hao Chen** *                       JHC@UST.HK
*Department of Computer Science and Engineering, Department of Chemical and Biological Engineering, Division of Life Science, and State Key Laboratory of Nervous System Disorders, Hong Kong University of Science and Technology, Hong Kong, China.*
*HKUST Shenzhen-Hong Kong Collaborative Innovation Research Institute, Futian, Shenzhen, China.*

**Editors:** Accepted for publication at MIDL 2026

## Abstract

Models based on human-understandable concepts have received extensive attention to improve model interpretability for trustworthy artificial intelligence in the field of medical image analysis. These methods can provide convincing explanations for model decisions but heavily rely on detailed annotations of predefined concepts. Consequently, they are ineffective when concepts or annotations are incomplete or of low quality. Although some methods can automatically discover novel and effective visual concepts instead of relying on predefined ones, or generate human-understandable concepts using large language models, they often deviate from medical diagnostic evidence and remain difficult to interpret. In this paper, we propose a concept complement bottleneck model for interpretable medical image diagnosis. Specifically, we use cross-attention modules to extract key image features related to the predefined textual concepts and employ independent concept adapters and bottleneck layers to distinguish concepts more effectively. Additionally, we devise a concept complement module to mine local concepts from the concept bank constructed using medical literature. The model jointly learns expert-annotated predefined concepts and automatically discovered ones to improve performance in concept detection and disease diagnosis. Comprehensive experiments demonstrate that our model outperforms state-of-the-art methods while providing diverse and interpretable explanations. Our code will be released on https://github.com/HongmeiWANG-HKUST/Concept-Complement-Bottleneck-Model.

**Keywords:** Explainable AI, medical image diagnosis, concept bottleneck model, concept discovery.

## 1. Introduction

Despite the impressive performance in Medical Image Analysis (MIA), AI models still face several challenges in clinical deployment. A critical challenge is that such black-box models lack transparency and interpretability throughout end-to-end training (Singla et al., 2023; Kasetty and Rajakumar, 2024). However, clinicians and patients need to understand how data are represented in the model's feature space and the evidence for model decisions

in this field. Such understanding fosters trust in the model's predictions and ultimately supports the deployment of AI in clinical practice (Kong et al., 2023; Liu et al., 2024).

Concept-based post-hoc models have attracted the most attention among explainable AI approaches because they use human-understandable textual or human-friendly visual concepts to explain model decisions (Gupta and Narayanan, 2024; Chauhan et al., 2023; Kim et al., 2023; Patrício et al., 2023; Hou et al., 2024). Some studies predict concepts that are present in images and have been densely annotated by clinicians to drive subsequent decisions. The Concept Bottleneck Model (CBM) (Koh et al., 2020) is a representative example, which learns concept scores by minimizing the loss of concept detection via a bottleneck layer. Variants of CBM have been widely explored for disease diagnosis using X-ray, ultrasound, and other imaging modalities (Chauhan et al., 2023; Marcinkevičs et al., 2024). Although CBM-based methods improve interpretability, they demand complete concept annotations during model training. In clinical practice, obtaining dense labels is extremely time-consuming and labor-intensive. Some approaches automatically discover visual concepts (Fang et al., 2020; Kong et al., 2024), yet these concepts rarely align with clinical terminology in a general domain. Leveraging Large Language Models (LLMs) or Vision-Language Models (VLMs), recent work automatically generates textual concepts for images (Shang et al., 2024; Yang et al., 2023). Nevertheless, such concepts only remain aligned with general-domain knowledge and often lack clinical relevance. Although these methods deliver competitive interpretability and accuracy, they either depend entirely on expert-annotated concepts or rely solely on automatically discovered ones for decision-making. The former demands dense concept labels yet still underperforms black-box models, whereas the latter often drifts away from authentic clinical reasoning. Moreover, existing concept-based approaches typically extract a single image feature map and reuse it for all concepts, ignoring inter-concept heterogeneity.

To tackle these limitations, we introduce the Concept Complement Bottleneck Model (CCBM). Unlike previous CBMs that only rely on fixed expert annotations or only discovering new concepts, CCBM automatically mines local textual concepts from medical literature, yielding higher accuracy and richer explanations without demanding extra manual labels. Our core contributions are as follows. First, the CCBM framework introduces a clinically-aligned concept bank derived from medical textbooks to ensure relevance and validity of disease diagnosis. Second, it departs from shared-feature architectures by employing per-concept processing with cross-attention, allowing each concept to have its own adapter and bottleneck layer to capture specific visual evidence effectively and achieve semantic decoupling. Additionally, a concept complement module dynamically merges local concepts with a global set to mitigate the information gap with black-box models inherent in static bottlenecks and enhance interpretability. Empirically, CCBM achieves notably performance improvements in disease diagnosis as well as concept detection on multiple medical imaging datasets, with verified interpretability through quantitative and qualitative analyses.

## 2. Methodology

The framework of our method is shown in Fig. 1. Our framework consists of offline concept bank construction and online end-to-end model training. In this section, we will introduce

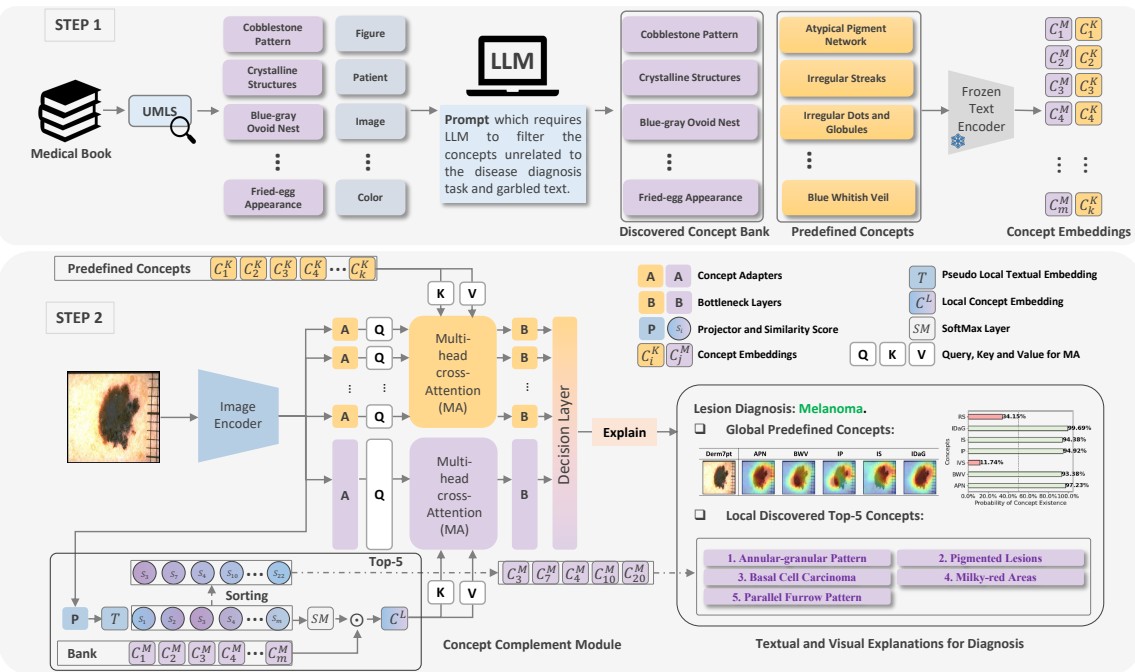

Figure 1: Concept complement bottleneck model. STEP 1: offline concept bank construction. STEP 2: online end-to-end concept complement Bottleneck Model Training. The CCBM uses multi-head attention modules to encode image features corresponding to each concept to guide the model decision. The first is to learn the visual features of global predefined concepts and the second is to discover effective local concepts in the textual concept bank via the concept complement module.

the construction of the concept bank from medical books and the detailed pipeline of CCBM to leverage global concepts to guide the discovery of local concepts from the bank.

### 2.1. Global Predefined Textual Concept Learning

The expert-annotated concepts are used for predefined global concept learning, and we encode the textual concepts and use an image encoder to extract the corresponding visual embeddings guided by the Multi-head cross-Attention (MA) module. The textual concept set is defined as $\mathcal{C}^{\mathcal{K}}$ which includes $k$ predefined labeled concepts, and the textual known concept encoder $E_T$ is used to encode the textual known concepts to the subspace with $d_c$ dimensions: $E_T : \mathcal{C}^{\mathcal{K}} \to \mathbb{R}^{d_c}$. Furthermore, the concept embeddings $C_i^K$ $(i = 1, 2, ..., k)$ will be used as the keys and values in the MA module for predefined concept learning.

### 2.2. Local Textual Concept Discovery

We construct a candidate concept bank where concepts are collected from medical books and filtered by an LLM. Then, CCBM can discover effective concepts from the concept bank

for each sample in the dataset. These textual concepts for a specific sample are defined as local concepts. We will introduce the pipeline to discover the local concepts in this section.

### 2.2.1. Concept Bank Construction

For local concept discovery, we first collect human-friendly textual medical concepts to build medical concept banks for different medical diagnosis tasks. As STEP 1 shown in Fig.1, for each task, we first extract concepts by Unified Medical Language System (UMLS) (Bodenreider, 2004; Singla et al., 2023) from medical books that are highly related to the specific task. However, there are some extracted concepts which are unrelated to the medical area, such as "Figure", "Image", "Criteria", etc. Therefore, we feed all extracted concepts into an LLM (GPT-4o (Hurst et al., 2024)) and give a suitable prompt to retain high-related medical concepts and remove low-related concepts. In this way, we build medical concept banks for diagnosis tasks easily. We formulate the concept bank with $m$ concepts as $\mathcal{C}^{\mathcal{M}}$. Similarly, using the same text encoder mentioned in the section 2.1, we can get the textual embeddings $C_i^M$ $(i = 1, 2, ..., m)$ for medical concepts.

### 2.2.2. Concept Complement Module

The concept complement module aims to project image features into the textual concept embedding subspace and learns local concepts from the concept bank. In this module, a linear projector $P$ projects basic image feature $\mathbf{f}$ to the textual concept embedding subspace and generates the pseudo local concept embedding $T$. As STEP 2 shown in Fig.1, the embedding $T$ will be used to calculate the cosine similarity with all textual concept embeddings in the concept bank. Next, these similarities will be activated by the $softmax$ layer to generate a probability vector to weigh all textual concept embeddings in the concept bank. For each image, the textual concepts with the larger weights will be considered as local concepts for the input image. We formulate the concept complement module:

$$s_j = (C_j^M \cdot T)/(\|C_j^M\|\|T\|), T = P(\mathbf{f}), \tag{1}$$

$$\mathbf{w} = softmax([s_1, s_2, ..., s_m]), \tag{2}$$

where $C_j^M$ is the $j$-th concept embedding in the concept bank, and $s_j$ means the cosine similarity of $C_j^M$. Then, we can obtain the local textual concept embedding for each image as $C^L = \mathbf{w} \cdot [C_1^M, C_2^M, ...C_m^M]$. For one image, based on the values of $\mathbf{w}$, we can obtain the importance scores of each concept in the concept bank.

## 2.3. Concept Complement Bottleneck Model

### 2.3.1. Dual Multi-Head Attention Branches for Concept Learning

Based on the branches of global predefined concept learning and local textual concept discovery, we can build the CCBM using the following steps. $E_T$ is the frozen text encoder to encode the textual concepts in the predefined concept set and concept bank. $E_I$ is the image encoder to map the input images $\mathcal{X}$ including $n$ samples from $n_c$ classes to the image feature space: $E_I : \mathcal{X} \rightarrow \mathbb{R}^d$, where $d$ represents the feature dimension. Intuitively, if one feature is used to calculate cross-attention effectively with all visual concept embeddings, the

image encoder needs to be strong enough to extract all crucial visual features for all different textual concepts. Here, we designate visual features as queries to actively retrieve relevant clinical descriptors from the concept bank (acting as keys/values). This design mimics the diagnostic process of clinicians who seek textual evidence based on observed visual findings, while effectively suppressing non-informative background artifacts by ensuring the attention is conditioned on local image patches. Considering that complex concepts are more difficult to learn, we leverage concept adapters to capture the differences in visual concepts. Hence, $k$ concept adapters $A_i^K (i = 1, ..., k)$ are set to extract specific concept features from the basic image features to be the queries $Q$ of the predefined concept MA module. Similarly, concept adapters $A^L$ extract visual features of the local concepts. In our setting, each concept adapter is set as a linear layer that maps the $d$-dimension feature space to the $d_c$-dimension subspace:

$$Q_i^K = A_i^K(E_I(\mathcal{X}))(i = 1, 2, ..., k), Q^L = A^L(E_I(\mathcal{X})) \tag{3}$$

where $Q_i^K$ is the $i$-th query in the MA module for predefined concept learning and $Q^L$ is the query in the MA module for local concept learning. The predefined concept embedding $\{C_1^K, C_2^K, ..., C_k^K\} = E_T(C^K)$ will be used as the keys and values in the MA module for predefined concept learning, and the local concept embedding $C^L$ will be used as the key and value in the MA module for local concept learning. We only formulate the single-head cross-attention mechanism instead of MA formulation:

$$Attn(Q, K) = softmax\left((QK^T)/(\sqrt{d_c})\right), Attn_w(Q, K, V) = Attn(Q, K)V \tag{4}$$

where $Attn$ is the attention map matrix whose elements represent the weights of different concept pairs, and $Attn_w$ is the weighted sum of all concept embeddings. Furthermore, we calculate the concept scores based on these attention scores. Different from the previous bottleneck models (Koh et al., 2020), which use one bottleneck layer to get all concept scores, we use an independent bottleneck layer to get the score for each concept. In our setting, each bottleneck layer was set as a linear projector layer, and the final concept score for each concept can be defined as $g_i = f_b(Attn_w(Q, K, V)_i)$.

### 2.3.2. Concept Complement Bottleneck Model for Medical Image Diagnosis

Although existing models based on concept bottleneck can maintain high classification performance while providing interpretability by training on concept detection tasks and classification tasks, the model performance needs to be balanced between concept detection and classification tasks. In addition, there is still a gap between explainable models and black-box models in classification tasks. In order to reduce this performance gap while maintaining model transparency, we propose a concept complement strategy to learn unknown concepts that are helpful for diagnosis. For our concept complement strategy, based on the concept scores, the CCBM makes decisions by a linear decision layer $f_d$:

$$\hat{Y} = f_d(\mathbf{g}) \in \mathbb{R}^c, \tag{5}$$

where $\mathbf{g} = [g_1, g_2, ..., g_k, g_{k+1}]$ is the concept score vector of the input image, and the first $k$ elements are the predefined concept scores and the last element indicates the total score of

discovered local concepts. $\hat{Y}$ is the final disease prediction. During the training process, we jointly train the model to perform well on the concept detection task and disease diagnosis task. In particular, we require model decisions to more explicitly depend on these concept scores to ensure model interpretability. Therefore, we leverage the cross-entropy loss for the classification task and the concept-learning loss for the concept detection task. The cross-entropy loss for disease diagnosis task and the multi-label classification cross-entropy loss for concept learning are defined as:

$$\mathcal{L}_{ce} = -\sum\nolimits_{i=1}^{N}\sum\nolimits_{j=1}^{C} y_{ij}\log(\hat{y}_{ij}), \tag{6}$$

$$\mathcal{L}_{cep} = -\sum\nolimits_{i=1}^{N}\sum\nolimits_{j=1}^{n_k}\left(c_{ij}\log(g'_{ij}) + (1 - c_{ij})\log(1 - g'_{ij})\right), \tag{7}$$

The total loss of the CCBM is:

$$\min_{\hat{Y},g}\left(\lambda\mathcal{L}_{ce}(\hat{Y}, Y) + \mathcal{L}_{cep}(g, C)\right), \tag{8}$$

where $\hat{Y}$ is the classification prediction of the model, $Y$ is the ground truth of image diagnosis, $C$ is the matrix of the ground truth of the concept detection task, and $\lambda$ is the hyperparameter to balance the two tasks.

## 3. Experiments

### 3.1. Experiment Settings

We conducted comprehensive experiments on two dermoscopic image datasets *Derm7pt* and *Skincon*, and an ultrasound breast image dataset *BrEaST*. **Derm7pt** (Kawahara et al., 2018) contains 1011 dermoscopic images, including 20 specific skin disease diagnoses and detailed labels of 7 clinical concepts based on the seven-point skin lesion malignancy checklist. We only considered 827 images in which the diagnosis belongs to *Nevus* and *Melanoma*. **Skincon** (Daneshjou et al., 2022) contains 3230 skin images of *Malignant*, *Benign*, and *Non-neoplastic* categories in the Fitzpatrick 17k dataset (Groh et al., 2021). We chose 22 concepts where there are at least 50 images containing the concept from 48 general medical clinical concepts densely annotated by two dermatologists. **BrEaST** (Pawłowska et al., 2024) is an ultrasound breast image dataset with detailed annotations via 7 concepts from BI-RADS descriptors, which contains 256 images with 3 different types of breast diagnosis, including *Benign*, *Malignant*, and *Normal*. We only use 254 abnormal breast images in our experiments. The details of our used concepts are shown in Tab. 1. The lists of medical concepts in the concept banks for skin disease diagnosis and ultrasound breast nodule diagnosis ($m = 73$ and $m = 84$, respectively) are provided in Tab. 4 of Appendix A. The prompts for LLM to filter concepts are provided in Tab. 5 of Appendix A.

As for dermoscopic image datasets *Derm7pt* and *Skincon*, we followed PCBM (Yuksekgonul et al., 2022) and choose the trained Inception-v3 model (Szegedy et al., 2015) as the image encoder to ensure fairness since the PCBM used the backbone, and we used pretrained ResNet50 as the image encoder for the *BrEaST* datatset. The frozen pre-trained ClinicalBERT (Alsentzer et al., 2019) was utilized as the text encoder. Additionally, the Adam optimizer was used for model training. We use the grid-search method to find the

best model hyperparameter $\lambda$ from $\{0.1, 0.2, 0.5, 1, 10\}$. To evaluate model performance, we use the Area Under Curve (AUC), ACCuracy (ACC), and F1-score as disease diagnosis evaluation metrics, and the first two metrics are also used to evaluate the concept detection tasks of *Derm7pt*, *Skincon*, and *BrEaST*. All of means and standard deviation results are obtained by five-fold cross-validation in our experiments.

Table 1: Global concept details in our experiments for three datasets.

| Dataset | Global Concepts |
|---------|-----------------|
| *Derm7pt* | Atypical Pigment Network (APN), Blue Whitish Veil (BWV), Irregular Vascular Structures (IVS), Irregular Pigmentation (IP), Irregular Streaks (IS), Irregular Dots and Globules (IDaD), Regression Structures (RS) |
| *Skincon* | PAPule (PAP), PLAque (PLA), PUStule (PUS), BULla (BUL), PATch (PAT), NODule (NOD), ULCer (ULC), CRUst (CRU), EROsion (ERO), ATRophy (ATR), EXUdate (EXU), TELangiectasia (TEL), SCALe (SCAL), SCAR (SCAR), FRIable (FRI), Dome-SHaped (DSH), Brown-Hyperpigmentation (BrH), White-Hypopigmentation (WhH), PURple (PUR), YELlow (YEL), BLAck (BLA) and ERYthema (ERY) |
| *BrEaST* | IRregular Shape (IRS), Not Circumscribed Margin (NCM), Hyperechoic or Heterogeneous Echogenicity (HoHE), Posterior Features (PF), Hyperechoic Halo (HH), CALcifications (CAL), Skin Thickening (ST) |

### 3.2. Comparison Algorithms

To verify the effectiveness and advancement of the model, we compare CCBM with the state-of-the-art methods, including CBM (Koh et al., 2020), PCBM (-H) (Yuksekgonul et al., 2022), an Ante-hoc Explainable Concept-based model (AEC) (Sarkar et al., 2022), Concept-Based interpretability using Vision-Language Models (CBVLM) (Patrício et al., 2023) and Energy-based CBM (ECBM) (Xu et al., 2024). Especially, we employed ResNet-50 (RN50) as the visual encoder and followed the GPT+CBM architecture. For the textual descriptions generated by the LLM, we directly used the original descriptors provided in the CBVLM official open-source repository to ensure fidelity to the original method. The concept extraction and reasoning were performed in a zero-shot setting during the inference phase. For detailed settings of other methods, we followed the recommended settings for other algorithms. We also tested the backbone models to evaluate the gap between our explainable model and black-box models. For methods that do not support concept detection, we use "N/A" to indicate invalid data.

### 3.3. Experimental Results and Analysis

#### 3.3.1. DISEASE DIAGNOSIS AND CONCEPT DETECTION

To verify the effectiveness of our model on the concept detection task and disease diagnosis task, we conducted extensive experiments and compared CCBM with the state-of-the-art methods on three datasets using multiple metrics. The five-fold cross-validation results of CCBM on these tasks are shown in Tab. 2. CCBM achieves outstanding performance for dermoscopic image analysis. For the *Derm7pt* dataset, CCBM achieves highest concept detection performance. It outperforms the comparison explainable methods with 93.15%

AUC, the 87.18% ACC, and 84.20% F1-score on the disease diagnosis and surpasses the black-box model on all metrics. Regarding the *Skincon* dataset, CCBM excels in the concept detection task on the AUC 80.51%, which is higher than the second with about 15% improvements. While maintaining strong concept detection performance, its AUC of disease diagnosis task reaches 85.15%, outperforming other competitors, and the 80.22% ACC is comparable with the best model, AEC. The black-box model underperforms on the dataset, with CCBM surpassing it by 5% on AUC, 3% on AUC, and 7% on F1-score. For ultrasound diagnosis and analysis, the results of the *BrEaST* dataset show that CCBM achieves the best performance on the concept detection task and the classification task on all evaluation metrics. The performance of CCBM is slightly better than other methods, with 88.89% AUC, 78.81% ACC, and 78.81% F1-sorce, especially outperforming the black-box model across all metrics. It is notable that the concept detection AUC improves about 4% on AUC as well as ACC compared with the second best model, representing a considerable improvement. Overall, CCBM discovers unknown concepts to complement the predefined known concepts to provide additional information for the model decision, and it outperforms other advanced explainable models and the compared black-box model on disease diagnosis while guaranteeing the best performance on concept detection tasks. Detailed concept detection results are provided in Fig. 5 in Appendix B.

Table 2: Quantitative results on disease diagnosis and concept detection tasks with comparison methods and black-box models. The results are shown as the mean ± std of the five-fold cross-validation experiment. "N/A" indicates invalid data.

| Dataset | Model | Disease Diagnosis | | | Concept Detection | |
|---|---|---|---|---|---|---|
| | | AUC (%) | ACC (%) | F1-score (%) | AUC (%) | ACC (%) |
| Derm7pt | PCBM | 81.32 ± 2.12 | 75.82 ± 2.00 | 71.10 ± 1.74 | N/A | N/A |
| | PCBM-H | 85.87 ± 1.53 | 78.60 ± 2.79 | 75.50 ± 3.09 | N/A | N/A |
| | CBVLM | 83.45 ± 3.59 | 84.13 ± 2.71 | 71.24 ± 2.54 | N/A | N/A |
| | ECBM | 75.03 ± 2.06 | 76.43 ± 2.47 | 73.50 ± 2.24 | 70.59 ± 2.60 | 78.85 ± 1.40 |
| | AEC | 91.27 ± 2.02 | 84.88 ± 2.05 | 80.99 ± 3.32 | 76.61 ± 1.61 | 75.30 ± 0.72 |
| | CBM | 92.88 ± 1.90 | 85.89 ± 1.92 | 82.18 ± 2.67 | 82.15 ± 2.68 | 80.00 ± 1.87 |
| | CCBM | **93.15 ± 3.68** | **87.18 ± 1.18** | **84.20 ± 4.90** | **83.07 ± 1.53** | **80.71 ± 0.87** |
| | Inception-v3 | 92.02 ± 2.53 | 86.46 ± 2.54 | 83.13 ± 3.31 | N/A | N/A |
| Skincon | PCBM | 69.06 ± 1.23 | 72.48 ± 1.56 | 39.55 ± 0.97 | N/A | N/A |
| | PCBM-H | 72.85 ± 1.66 | 68.42 ± 3.07 | 53.33 ± 3.38 | N/A | N/A |
| | CBVLM | 71.34 ± 2.16 | 70.74 ± 1.11 | 66.15 ± 1.50 | N/A | N/A |
| | ECBM | 68.04 ± 1.68 | 79.16 ± 1.06 | 61.39 ± 2.29 | 64.64 ± 0.97 | 90.94 ± 0.20 |
| | AEC | 83.86 ± 0.61 | **80.24 ± 0.52** | 63.89 ± 1.63 | 58.64 ± 0.90 | 90.64 ± 0.10 |
| | CBM | 80.01 ± 1.25 | 78.42 ± 1.31 | 60.57 ± 3.04 | 62.14 ± 1.37 | 89.32 ± 0.14 |
| | CCBM | **85.15 ± 1.45** | 80.22 ± 0.90 | **67.08 ± 0.62** | **80.51 ± 0.51** | **91.32 ± 0.14** |
| | Inception-v3 | 79.92 ± 1.48 | 77.52 ± 1.47 | 59.86 ± 2.78 | N/A | N/A |
| BrEaST | PCBM | 75.41 ± 5.74 | 68.43 ± 7.64 | 64.79 ± 8.28 | N/A | N/A |
| | PCBM-H | 79.63 ± 2.95 | 70.02 ± 3.87 | 67.73 ± 3.81 | N/A | N/A |
| | ECBM | 68.09 ± 7.50 | 68.83 ± 6.10 | 66.76 ± 6.24 | 59.20 ± 1.93 | 77.94 ± 1.65 |
| | AEC | 82.80 ± 3.32 | 72.40 ± 3.39 | 68.60 ± 3.51 | 69.71 ± 1.52 | 77.82 ± 1.70 |
| | CBM | 87.42 ± 4.27 | 77.21 ± 8.62 | 76.29 ± 8.31 | 70.76 ± 1.38 | 77.49 ± 1.43 |
| | CCBM | **88.89 ± 3.67** | **78.81 ± 6.60** | **76.73 ± 7.12** | **74.31 ± 3.32** | **81.03 ± 2.16** |
| | ResNet50 | 86.97 ± 6.14 | 77.61 ± 6.23 | 76.39 ± 6.09 | N/A | N/A |

### 3.3.2. Ablation Study

To further explore the impact of the discovered local concepts and the effectiveness of independent concept adapters as well as bottleneck layers, we conducted well-designed ablation experiments on the *Derm7pt*, *Skincon*, and *BrEaST* datasets. The results are summarized in Tab. 3, where "BASE" defines the model which removes local concept branch from CCBM to evaluate the effectiveness of the whole local concept discovery branch. "W/O-AB' means that the concept adapters and bottleneck layers are replaced by one linear layer, respectively, compared with standard CCBM. "R-A" means that the visual features from different adapters are replaced by repeated visual features from one adapter.

Comparing the performance of "BASE" and other competitors in Tab. 2, the CCBM's performance remains competitive with other explainable methods even when the local concept learning branch is removed, especially for the concept detection tasks. Moreover, with the local concept learning, the performance of CCBM on the two tasks is markedly improved, which means the discovered local concepts revise the weight distribution of predefined concepts and provide more information to improve model predictions in conjunction with the existing predefined concepts. Further, we can see that the results of the "R-A" are worse than the standard CCBM, which means the independent concept adapters are effective to improve model performance. The results of the "W/O-AB" setting are generally worse than those of the "R-A", which further illustrates the effectiveness of independent adapters and bottleneck layers. For the *Derm7pt* and *BrEaST*, which have limited predefined global concepts, there is potential to find more valuable concepts from the related medical domain. Hence, more performance improvements can be observed on these datasets. Even for the dataset with enough predefined concepts, the *Skincon*, CCBM can further improve the accuracy for image classification and concept detection.

## 3.4. Explanability Analysis

### 3.4.1. Inference-time Intervention for Faithfulness

Faithfulness indicates that the model explanations could faithfully elucidate the model decision. In the inference-time intervention experiments, we artificially modified partial concept predictions to observe the resulting changes in the final model decisions, allowing us to assess the effectiveness of the concept explanations. Specifically, we established a set of thresholds for concept scores during model inference, where any concept scores surpassing the threshold are reset to zero. Here, we implemented a fixed thresholding strategy. For each fold, we first determined the maximum and minimum values of the concept logits in the test set. The range between these values is then equally divided into ten intervals, with the boundary points from smallest to largest serving as thresholds $t_1$ to $t_{10}$. To obtain the final intervention results, we sequentially set the concept logits as zeros based on these thresholds within each fold and recorded the corresponding metrics at each threshold. Then, the mean and variance of metrics are obtained across the five folds for each threshold.

We present the diagnosis outcomes inferred using the intervened concept scores. Fig. 2 shows that model disease diagnosis performance is notably impacted by the intervention of concept scores. For these three datasets, the AUC, ACC, and F1-score generally exhibit considerable improvements as the intervention threshold increases. These intervention experiments underscore the model's heavy reliance on predicted concept scores and affirm the

Table 3: Results of ablation experiments. The "BASE" defines the CCBM model without the local concept branch, the "W/O-AB" means that the concept adapters and bottleneck layers are replaced by linear layers, respectively. The "R-A" means that the different visual features from adapters are replaced by repeated visual features from one adapter in CCBM. The results are shown as the mean ± std of the five-fold cross-validation experiments.

| Dataset | Type | Disease Diagnosis | | | Concept Detection | |
|---|---|---|---|---|---|---|
| | | AUC (%) | ACC (%) | F1-score (%) | AUC (%) | ACC (%) |
| Derm7pt | BASE | 91.68 ± 1.76 | 83.92 ± 2.91 | 81.52 ± 3.22 | 82.78 ± 2.28 | 80.30 ± 1.78 |
| | W/O-AB | 91.44 ± 1.38 | 83.44 ± 1.93 | 78.34 ± 3.85 | 75.37 ± 1.65 | 77.01 ± 2.33 |
| | R-A | 91.29 ± 1.29 | 84.77 ± 1.85 | 80.88 ± 2.78 | 77.44 ± 1.55 | 77.04 ± 1.54 |
| | CCBM | **93.15 ± 3.68** | **87.18 ± 1.18** | **84.20 ± 4.90** | **83.07 ± 1.53** | **80.71 ± 0.87** |
| Skincon | BASE | 84.41 ± 1.20 | **80.25 ± 1.20** | **67.34 ± 2.72** | 79.27 ± 0.75 | 91.31 ± 0.20 |
| | W/O-AB | 82.93 ± 1.79 | 78.02 ± 1.86 | 63.15 ± 2.61 | 72.84 ± 1.64 | 90.72 ± 0.37 |
| | R-A | 82.94 ± 1.42 | 78.79 ± 1.10 | 64.45 ± 1.79 | 74.08 ± 2.72 | 90.46 ± 0.32 |
| | CCBM | **85.15 ± 1.45** | 80.22 ± 0.90 | 67.08 ± 0.62 | **80.51 ± 0.51** | **91.32 ± 0.14** |
| BrEaST | BASE | 88.16 ± 4.52 | 73.57 ± 14.07 | 72.79 ± 14.17 | 66.49 ± 4.72 | 79.52 ± 4.26 |
| | W/O-AB | 86.55 ± 3.52 | 74.41 ± 6.94 | 71.99 ± 8.83 | 71.27 ± 6.89 | 80.51 ± 2.20 |
| | R-A | 87.47 ± 3.54 | 78.00 ± 4.39 | **76.78 ± 3.62** | 69.73 ± 4.30 | **81.03 ± 1.61** |
| | CCBM | **88.89 ± 3.67** | **78.81 ± 6.60** | 76.73 ± 7.12 | **74.31 ± 3.32** | 81.03 ± 2.16 |

faithfulness of the explanations provided. For the datasets with a small size, the effect of interference on the data results of different folds is more obvious, so there is a larger variance observed in the results from the *Derm7pt* and *BrEaST* datasets. The inference-time intervention experiments for each global concept are shown in Fig. 6 in Appendix C where the concept scores for specific concept are set to zeros to evaluate the importance of each concept. We can see that only some concepts are very important to model decision, while interfering with some concepts only results in a slight decrease in performance. However, comparing the decrease pattern shown in Fig. 2, it can be observed that the synergistic effect between concepts has a huge impact on model performance.

### 3.4.2. Visual and Textual Explanations for Plausibility

As shown in Fig. 3, we showcase the explanations for two images from the *Derm7pt* and *BrEaST* to illustrate the application on melanoma diagnosis and breast nodule diagnosis tasks. We can see that CCBM can capture the areas related the specific predefined concepts and generate the corresponding concept activation maps using Grad-CAM (Selvaraju et al., 2020), where the concept labels were used to activate the map from the bottleneck layers. Additionally, we can also provide the detailed concept scores and generated textual summaries using a template to report the concepts in the image and the top-k discovered local concepts. Specifically, the spatial overlap of diagnostic concepts for skin diseases can be observed in the activation maps for the melanoma example, which is consistent with real clinical situations. For the explanations of breast nodule case, our model highlighted the correct concept-related regions. "NCM" and "HH" focus on the edge of the lesion, and the activation map for "IRS" highlights the most irregular region of the lesion.

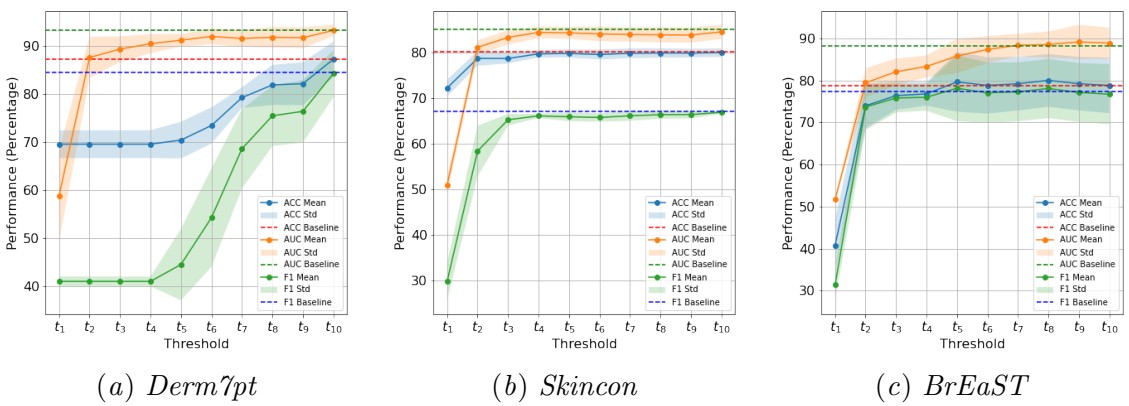

Figure 2: Inference-time intervention results on (a) *Derm7pt*, (b) *Skincon*, and (c) *BrEaST*. The $x$-axis represents the thresholds ($t_1 < t_2 < ... < t_{10}$), and the $y$-axis indicates the percentage of diagnosis performance metrics after intervention.

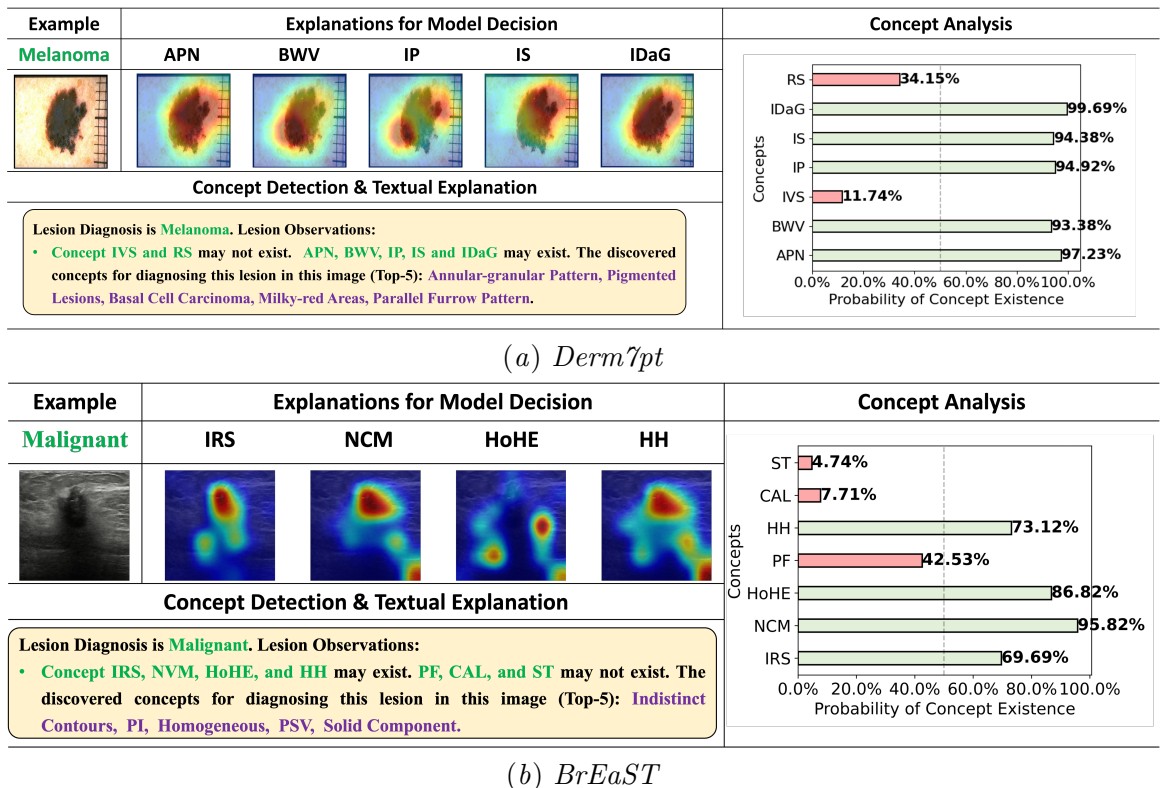

Figure 3: Visual and textual explanations of images in (a) *Derm7pt* and (b) *BrEaST*, and the discovered concept analysis for the datasets.

We visualized the distribution of the discovered top-1 concepts of training and testing samples as shown in Fig. 4. It can be found that the first four concepts of the training data and the test data are the same, although their rankings are not completely consistent. These concepts are highly related to the characteristics of melanoma, which reflects that the discovered concepts can indicate key features to distinguish the categories and guide model predictions. With the local branch, the weights of the concepts ("Irregular Vascular Structures (IVS)" and "Irregular Dots and Globules") decreased obviously. The discovered concepts, "Red Dots" and "Corona Vessel", are highly related to the above concepts semantically, but are not the same. This means that CCBM discovers text concepts that are more beneficial for the classification tasks. Additionally, "Annular-granular Pattern" and "Radial Streaming" are also highly specific dermoscopic signs of melanoma. Specifically, the concept "Annular-granular Pattern" is highly indicative of lentigo maligna, which is a type of melanoma in situ. The "Radial Streaming" indicates focal irregular streaking or peripheral linear projections.

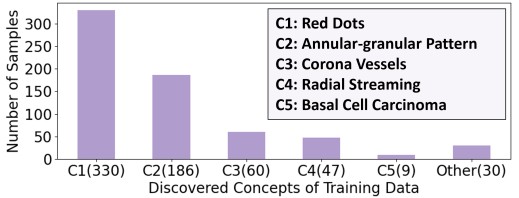
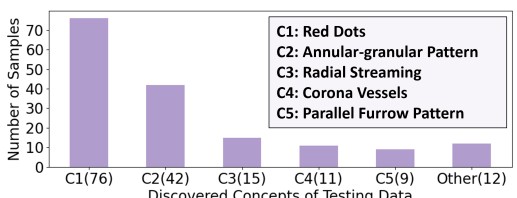

($a$) Discovered Concepts from Training          ($b$) Discovered Concepts from Testing

Figure 4: The statistical results of the top-1 discovered concepts of (a) the training samples and (b) the testing samples for the *Derm7pt* dataset.

## 4. Conclusion

In this paper, we introduce a concept complement bottleneck model for interpretable medical image diagnosis. By incorporating concept adapters with visual-text concept cross-attention modules, CCBM discovers important concepts from a concept bank while simultaneously learning predefined concepts for disease diagnosis. Comprehensive experiments demonstrate that CCBM achieves superior classification performance in both concept detection and disease diagnosis tasks, providing faithful and interpretable explanations. Despite the promising performance, our method is currently limited by the scale of the knowledge bases and datasets used for validation. Future research will focus on extending this framework to larger-scale datasets and exploring the complex interdependencies between global expert knowledge and local discovered clinical concepts to further enhance the model's robustness and generalizability.

## 5. Acknowledgments

This work was supported by the Hong Kong Innovation and Technology Fund (Project No. MHP/002/22), HKUST (Project No. FS111), and the Research Grants Council of the Hong Kong Special Administrative Region, China (Project Reference Number: T45-401/22-N).

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

## Appendix A. Concept List in Concept Bank and LLM Prompts

## Appendix B. Details of concept detection

Fig. 5 displays the fine-grained evaluation results of CCBM and other competitors on the concept detection task. CCBM achieves the highest average AUCs for the three datasets. Particularly, our CCBM demonstrates strong performance across most concepts, while CBM and other models tend to wrongly classify the concepts which have fewer positive samples. For example, for the *Skincon*, the results of "BUL", "PUR", "ATR", and "BLA" are representative. While the number of their positive samples is smaller than one hundred, the concept accuracy of CCBM is higher by 10% to 20% than the sub-optimal models. It indicates that CCBM learns each concept more effectively and fairly, benefiting from the concept adapters and independent bottleneck layers.

## Appendix C. Inference-time Intervention for Faithfulness on Each Concept

Table 4: The $m$ candidate local concept list in concept bank.

| Dataset | Concept List in Local Concept Bank |
|---------|-------------------------------------|
| Derm7pt & Skincon | melanoma, nevus, dermoscopy, pigmented lesions, asymmetry, irregular borders, globules, streaks, reticular pattern, vascular structures, blue-white veil, brown pigmentation, black color, red dots, white areas, homogenous color, network lines, atypical vessels, regression structures, spitzoid pattern, milky-red areas, blue-gray granules, pseudopods, radial streaming, leaf-like areas, lacunae, cobblestone pattern, honeycomb pattern, crystalline structures, blue nevus, seborrheic keratosis, basal cell carcinoma, squamous cell carcinoma, actinic keratosis, dermatofibroma, hypopigmented, hyperpigmented, ulceration, nodular, verrucous, polymorphous vessels, arborizing telangiectasia, hairpin vessels, glomerular vessels, starburst pattern, parallel furrow pattern, fibrillar pattern, shiny white lines, chrysalis structures, blue-gray ovoid nests, comma vessels, corona vessels, dotted vessels, serpentine vessels, structureless zones, milia-like cysts, comedo-like openings, central hyperkeratosis, peripheral rim, blue-black pigmentation, red-blue lagoons, white reticular network, gray-blue dots, rhomboid structures, annular-granular pattern, fried-egg appearance, targetoid pattern, strawberry pattern, blue-gray blotches, black lamellae, peppering, blue-gray veil, negative network |
| BrEaST | elastography, strain elastography, shear wave elastography, contrast-enhanced ultrasound, CEUS, harmonic imaging, 3D reconstruction, ductography, hypoechoic, hyperechoic, anechoic, irregular shape, oval, lobulated, spherical, smooth margins, irregular margins, spiculated margins, microlobulated, angular margins, circumscribed, indistinct contours, homogeneous, heterogeneous, posterior enhancement, posterior shadowing, acoustic shadowing, edge shadowing, parallel orientation, non-parallel orientation, taller-than-wide, wider-than-tall, vascularity, hypervascular, avascular, intranodular vascularization, peritumoral vascularity, intratumoral vascularity, neovascularization, tumor angiogenesis, vascular resistance index, RI, pulsatility index, PI, PSV, solid component, complex cyst, intracystic mass, cystic wall, septations, calcifications, microcalcifications, ductal extension, ductal ectasia, ductal wall thickening, intraductal mass, papilloma, radial scar, fibroadenoma, phyllodes tumor, invasive ductal carcinoma, lobular cancer, ductal carcinoma in situ, DCIS, benign lesions, malignant tumors, malignant transformation, axillary lymph nodes, lymph node metastasis, BI-RADS classification, US-guided biopsy, core needle biopsy, FNAB, US probe, MHz frequency, breast parenchyma, Cooper's ligaments, subcutaneous fat, glandular tissue, retromammary space, nipple inversion, skin thickening, tumor size, tumor margins |

Table 5: The prompts of GPT-4o (Hurst et al., 2024) for filtering the concepts extracted from medical books using UMLS.

| Dataset | Prompt |
|---|---|
| Derm7pt & Skincon | Please filter the given list of concepts to select terms or phrases related to the diagnosis of skin diseases, removing irrelevant concepts and those with incorrect formatting. Please output the list of retained terms. |
| BrEaST | Please filter the given list of concepts to select terms or phrases related to the diagnosis of breast nodules based on ultrasound images, removing irrelevant concepts and those with incorrect formatting. Please output the list of retained terms. |

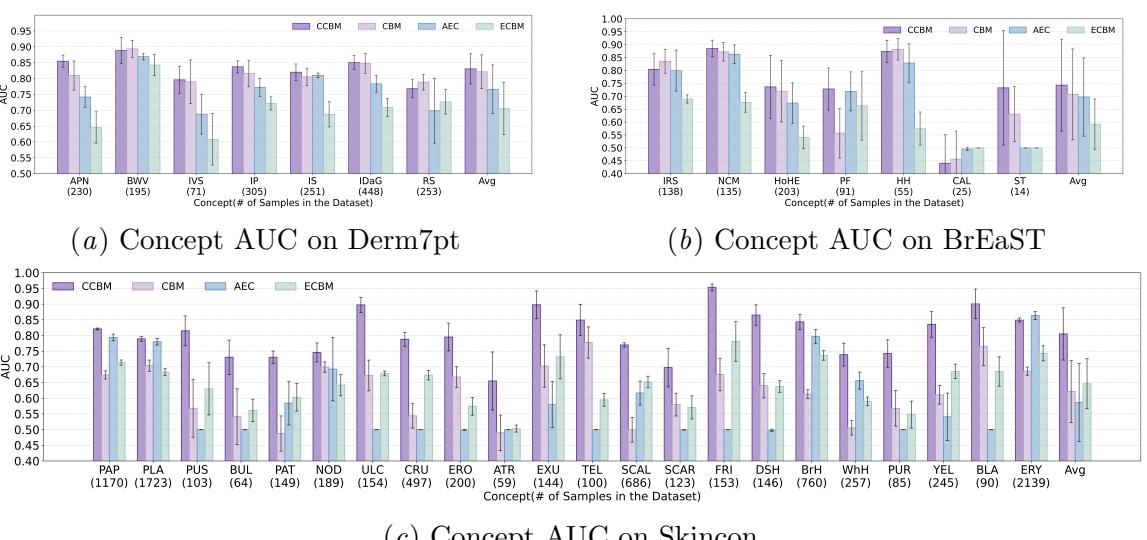

($a$) Concept AUC on Derm7pt          ($b$) Concept AUC on BrEaST

($c$) Concept AUC on Skincon

Figure 5: The fine-grained results of the concept detection task on the (a) *Derm7pt*, (b) *BrEaST*, and (c) *Skincon* datasets. The results are the means and stds of the five-fold cross-validation experiments. The "Avg" is the mean and std of the concept AUCs over all the concepts.

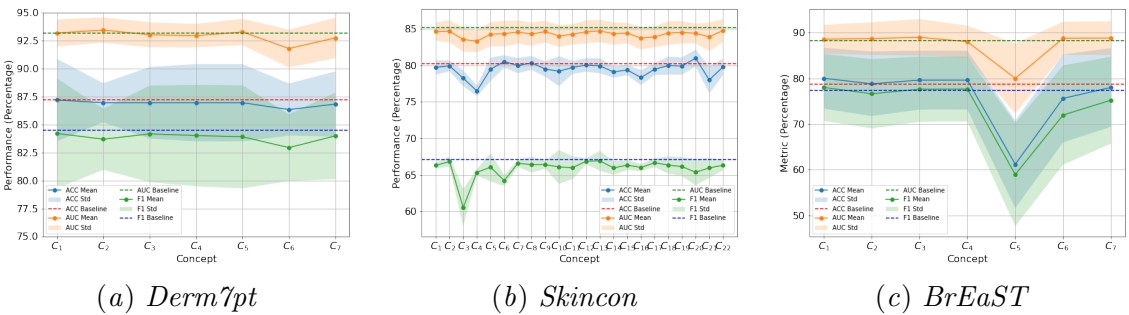

(a) Derm7pt       (b) Skincon       (c) BrEaST

Figure 6: Inference-time intervention results for each global concept on the (a) Derm7pt, (b) Skincon, and (c) BrEaST. The x-axis represents the concepts, and the order is consistent with the textual concepts in the Tab. 1. The y-axis indicates the percentage of diagnosis performance metrics after intervention.

