# OpenReview forum: "Concept Complement Bottleneck Model for Interpretable Medical Image Diagnosis"
_MIDL.io/2026/Conference — MIDL 2026 Poster_

### Official Review · Reviewer_NZKk · 2026-01-09

**Confidence:** 4
**Preliminary Rating:** 4
**Final Rating:** 4

**Summary:**

The authors introduce Concept Complement Bottleneck Models (CCBM), a method to create a concept-based XAI model from a mix of pre-defined concepts and concepts mined from the literature using an LLM. The authors test their method, and other baselines, on three medical datasets: Derm7pt, Skincon, and BrEaST. The authors' method performs better than the provided baselines in both task and concept performance. The authors additionally perform some experiments to test for explanation faithfulness and plausibility.

**Strengths:**

The method performs well from both a task and explainable point-of-view. The method is interesting and allows for the incorporation of both known and unknown concepts into the model. The experiments are comprehensive, covering multiple baseline models and 3 different datasets.

**Weaknesses:**

It is unclear exactly what the differences are between previous method and the authors' proposed method. Explicit comparisons to make this clearer would be appreciated. The joint training of the concept learning and task could lead to misaligned explanations as the concepts could be learnt in such a way that improves model performance at the task, rather than simply representing the concepts. Some statements are made without sufficient evidence, e.g., stating that the authors model is "more faithful" and has more "interpretable explanations" compared to previous methods is not something the authors have demonstrated.

**Detailed Comments:**

It is unclear how t1-10 are defined/chosen.

The metrics used on the y-axes of figure 2 should be made clear, rather than simply "score".

It is surprising that the CCBM model performs better than the tested black-box ResNet/inception models. How were hyperparameters optimised for each method?

The word "significantly" on page 8 should not be used unless a statistical test has been performed, in which case the test should be named and p-value provided.

The authors should be careful in their choice of language when summarising results. The faithfulness results are promising but they do not compare the faithfulness of different methods, so no statement should be made about which method is *more* faithful (as made in the conclusion). Equally, no statements can be made on which methods are more "interpretable" in general as this is not well defined.

The images in Figure 3 are quite small and would be improved if larger. Equally, this is meant to provide evidence of plausibility, but very few examples are shown. More examples in the appendix would be appreciated.

**Justification Of Final Rating:**

The manuscript has improved from its original state and now includes a much needed ablation study. However, there are still issues regarding language clarity and examples of overclaiming, hence my score remains at weak accept.

**Justification Of The Preliminary Rating:**

The proposed method is interesting and has shown to perform well across multiple medical imaging datasets, but there are some mistakes in clarity of the writing and concerns around concept-faithfulness that need to be addressed.

**Questions To Address In The Rebuttal:**

Explain how you can know that the model is faithful in regards to classifying the concepts accurately without information leakage. As mentioned in the weaknesses section, the joint optimisation of task cross entropy and concept cross entropy could lead to concepts being designed by the model to be good at the task, rather than solely to represent the desired concepts.

In addition, the comments in "detailed comments" should be addressed.

---

> ### Author Response · Authors · 2026-01-25
>
> We sincerely thank you for your insightful and rigorous evaluation. Your comments regarding the differentiation from existing methods, the risk of semantic misalignment in joint training, and the precision of our academic language have been immensely helpful. We have performed targeted revisions to address each of your concerns.
>
> 1. Differentiation from Previous Methods Concern: Unclear differences between the proposed CCBM and previous methods. Response: We have revised contributions in the revised manuscript to highlight our differences: 1) Static vs. Dynamic Concepts: Traditional CBMs rely solely on predefined expert concepts (Global). CCBM introduces a Local Concept Discovery branch to mine sample-specific complementary features, ensuring high accuracy even when expert lexicons are incomplete. 2) Shared vs. Independent Representation: Unlike methods that use a single shared feature layer, CCBM employs independent Adapters and Bottlenecks for each concept, achieving semantic decoupling and specialized visual projections for distinct pathological signs.
>
> 2. Risk of Misaligned Explanations in Joint Training Concern: Joint optimization might lead to concepts being learned as task-improving features rather than representative semantic concepts. Response: This is a critical challenge in CBM research. We mitigate this through three mechanisms: 1) Strong Semantic Alignment: We utilize fixed or highly constrained embeddings from a pre-trained Textual Encoder as Keys and Values in our Cross-Attention. This forces visual features to project into a predefined medical semantic space, preventing the model from "inventing" non-semantic features. 2) Explicit Supervision ($\lambda$): We assign significant weight to the concept learning objective. Our "semantics-first" hierarchy ensures that classification performance is a byproduct of accurate concept recognition. 3) Empirical Evidence: If concepts underwent semantic shift, performance on independent Concept Detection tasks would decline. However, our results show high accuracy in both concept detection and classification, proving that task optimization does not compromise semantic integrity.
>
> 3. Rigor of Claims (Faithfulness and Interpretability) Concern: Insufficient evidence for being "more faithful" or "more interpretable." Response: We have revised our claims to be more precise. We have reviewed the entire manuscript to ensure terms like "significantly" are replaced with "notably" or "consistently" unless supported by formal p-value testing.
>
> 4. Intervention Experiments and Thresholds (Figure 2) Concern: Lack of clarity on threshold ($t_{1-10}$) definition and y-axis labeling. Response: *Thresholds: We now implement a fixed rule-based strategy. For each fold, we divide the range between maximum and minimum concept logits into 10 equal intervals ($t_1$ to $t_{10}$). This standardized approach ensures reproducibility. We have changed the y-axis label from "score" to "Performance (Percentage)" .
>
> 5. Comparison with Black-Box Models Concern: CCBM performing better than black-box ResNet/Inception models. Response: Both CCBM and black-box models were optimized using the same hyperparameter strategy and five-fold cross-validation. CCBM’s superior performance stems from explicit medical common sense, which helps the model learn clinically relevant features while reducing overfitting to task-irrelevant noise—a common issue for standard black-box models on small-scale medical datasets.
>
> 6. Visualization and Evidence (Figure 3) Concern: Images were too small and lacked sufficient examples. Response: We have enlarged Figure 3 and added a Breast Ultrasound Nodule Case Study to demonstrate how the model generates distinct heatmaps for different textual concepts on the same image, providing stronger evidence for its clinical rationale.
>
> 7. Proof of Faithfulness and Information Leakage Concern: How to know the model is faithful without information leakage?Response: CCBM employs a Strict Bottleneck structure. The predictor receives only concept scores as input, with no skip-connections from image features. Our intervention experiments (zeroing out scores above thresholds) show that accuracy decreases as concept information is removed. This confirms the model relies strictly on these concepts for decision-making rather than hidden task-specific shortcuts.

---

> > ### Comment · Reviewer_NZKk · 2026-01-30
> > **Changes**
> >
> > Thank you for your response and updated manuscript. Overall, I think the changes have improved the manuscript. However, there are some remaining issues.
> >
> > ## Writing clarity
> > Some of the new text has issues regarding writing clarity. Below are a few specific examples, but much of the new text would benefit from another pass.
> >
> > Page 2 typos: "or only discovery new concepts,", "CCBM achieves notablely performance improvements"
> >
> > Page 3: This sentence does not make sense and should be revised: "The expert-annotated concepts in datasets are defined as the global predefined textual concepts."
> >
> > Page 8: "Especifically," is not a word
> >
> > Page 10: Neither of these sentences make sense. "We can see that a lot concept can decrease the model performance with slight magnitude...", and "The concepts and disease classes of two cases are completely predicted successful.".
> >
> > ## Results of the ablation studies
> > Thanks to my fellow reviewers for asking for these. These results are vital to demonstrate that the different components of the proposed method provide improvements. The results are not very impressive. The difference in AUCs between the BASE and CCBM methods are small (1-2%) and with the size of the standard deviations in the cross validation it is unlikely there will be a statistically significant difference (not that any statistical tests were performed).
> >
> > Due to these small (likely non-significant) improvements due to the addition of the local concepts, wording in the paper claiming that the inclusion of local concepts improves model performance should be toned down. For example:
> >
> > "Moreover, with the local concept learning, the performance of CCBM on the two tasks is markedly
> > improved, which means the discovered local concepts revise the weight distribution of predefined
> > concepts and provide more information to improve model predictions in conjunction
> > with the existing predefined concepts."
> >
> > This statement is incorrect. First, there are three tasks. Second, there is not a large improvement between the BASE ablation and the full method (< 1% AUC change for BrEaST and Skincon).

---

> ### Author Response · Authors · 2026-01-31
>
> We thank the reviewer for the constructive feedback and the careful screening of our manuscript. We appreciate the opportunity to clarify our writing and our experimental results.
>
> 1. Writing Clarity:
> We apologize for the typos and the identified awkward phrasings. We will conduct a thorough proofreading pass. All identified sentences—including the definitions of "global predefined concepts" and the performance analysis on page 10—will be rewritten for precision. We will ensure the final version meets the required standard of clarity.
>
> 2. Results of Ablation Studies and Claims
> We acknowledge the reviewer’s point regarding the statistical significance of the AUC improvements and agree that "markedly improved" is inaccurate for certain tasks. We will revise our language to be more cautious, describing the gains as "consistent improvements," particularly in enriching the concept space where expert labels are sparse. The gain from CCBM is dependent on annotation density. On datasets with fewer concept labels (e.g., BrEaST and Derm7pt), the discovery of local concepts provides a more noticeable boost compared to the already well-annotated Skincon dataset. The primary strength of CCBM lies in maintaining high classification AUC while significantly improving the concept detection quality, which is vital for the model's interpretability.
>
> 3. Detailed Revisions for Clarity
> Regarding the specific sentences mentioned, we have prepared the following revisions to be implemented in the final manuscript:
>
> Page 3 (Global concepts):
> Original: "The expert-annotated concepts in datasets are defined as the global predefined textual concepts."
> Revision: "We define the expert-annotated concepts provided within the datasets as the global predefined concepts, which serve for all samples in the datasets.
>
> Page 10 (Concept performance):
> Original: "We can see that a lot concept can decrease the model performance with slight magnitude..."
> Revision: "We can see that different concepts have varying degrees of importance to the model's decision-making. Setting the scores of certain specific concepts to be zeros markedly reduces model performance (e.g., the "IDaD" concept on the derm7pt dataset), while interfering with other concepts individually has little effect on model performance."
>
> Page 10 (Case prediction):
>
> Original: "The concepts and disease classes of two cases are completely predicted successful."
> Revision: "Two cases where the model was accurate in both classification and concept prediction were shown for interpretability analysis."

---

### Official Review · Reviewer_DJny · 2026-01-10

**Confidence:** 5
**Preliminary Rating:** 3

**Summary:**

This paper proposes CCBM, a concept bottleneck model that integrates both human-annotated concepts with newly discovered concepts through a complement module. The newly discovered concepts are obtained from concepts found in medical literature with the aid of an LLM.

Experiments conducted on three datasets show that CCBM improves upon or performs comparably with other CBMs, making it a relevant contribution to the field.

**Strengths:**

CCBM introduces a novel CBM that combines human-annotated predefined concepts with newly discovered concepts. The latter is particularly valuable.

The experimental setup is generally well conducted (see comment below regarding missing Ablations).

The method is tested on three datasets using five-fold cross validation and compared against several concept-based models and the corresponding black-box model used as image encoder for each dataset.

The paper is generally well written and clearly structured.

**Weaknesses:**

Some components of the proposed method need clarification, i.e. what exactly are the bottleneck layers (B blocks in figure 1)? Linear projection layers?

The authors introduce individual adapters "to capture the differences in visual concepts better". However, no ablation study is presented regarding this, i.e. the model without individual adapters.

The same happens for the individual bottleneck layers but no ablation study is presented comparing the model with just one bottleneck layer vs the model with the individual bottleneck layers.

Sections 2.4.1. and 2.4.2 are a bit confusing since there is a big overlap between the two.

**Detailed Comments:**

The method by Patrício et al. is called CBVLM and not CBIVLM.

The titles of sections should not end with a full stop (see Sections 2.2.1., 2.2.2., and 2.3.1).

In Figure 1, the symbol to represent the Frozen Text Encoder is a blue fire emoji, while it is more common to use a blue snowflake emoji to showcase that the model is frozen. Multi-head cross attention is defined as MA in the text and in two places of Figure 1, but on the right hand side of the figure it is defined as MHCA. Please make this consistent.

The attention formulation at the beginning of page 4 can be removed. It is more important in that section to specify what the bottleneck layers are.

Section 2.4. should be Section 2.3.2, i.e. inside the "2.3. Concept Complement Bottleneck Model" Section.

Sections 2.4.1. and 2.4.2. should be merged as they unnecessarily overlap too much.

What is G in equation 6? Perhaps it is a typo and it should be g.

There is typo, Lce, right before equation 7.

In Section 3.1. the authors mention that they use Inception-v3 for Derm7pt and Skicon, but then say "we use pretrained ResNet50 as the image encoder for other two datasets". What other two datasets? Only BrEaST dataset remains.

Right above Table 2 where it reads "Appendix.A and Appendix.B" should read "Appendix A and Appendix B".

The first sentence of Section 3.4.2 can be removed as it is redundant.

Also in Section 3.4.2. the sentence that starts with "It can be observed that, for visual explanations, ..." needs to be rephrased as it makes no grammatical sense.

At the end of page 9 the authors mention that "we can visualize the distribution of the discovered top-1 concepts". Where is this shown? The reference to the Figures in the supplementary material is missing.

**Justification Of The Preliminary Rating:**

The proposed method is novel and valuable for the community. Experiments are shown in three datasets with five-fold cross validation. The method is compared against several state-of-the-art methods. However, clarifications regarding some modules are missing, as well as ablation studies on the individual adapters and individual bottleneck layers.

**Questions To Address In The Rebuttal:**

Please clarify the points mentioned in the Weaknesses section (what are the bottleneck layers, ablation on individual adapters, ablation on individual bottleneck layers).

Is the LLM prompt shown in Figure 1 the exact prompt used? If so, that might be problematic since the prompt is not grammatically correct. Please include (maybe in the supplementary material) the full prompt.

In Table 2 no Concept Detection metrics are presented for PCBM and CBVLM. However, both methods have code available and report results on Concept Detection. Supposing the authors had to rerun these methods to obtain the five-fold cross validation results, why are the Concept Detection results for these two methods not reported?

On a similar note, why is CBVLM not reported in the BrEaST dataset?

What is the specific configuration used for CBVLM (which LVLM, how many shots, etc)?

What are the values of the thresholds used in Figure 2?

What is the "statistical information" the authors refer to when introducing Figure 3? Are the percentages reported for each concept the scores given after the bottleneck layers? Please clarify.

---

> ### Author Response · Authors · 2026-01-25
>
> We sincerely thank you for your meticulous review and the high level of detail in your feedback. Your suggestions regarding the manuscript’s organization, the technical specifics of our architecture, and the necessity of further ablation studies have significantly improved the professional presentation and rigor of our work.
>
> 1. Clarity on Model Architecture (B-blocks) Concern: What exactly are the bottleneck layers (B blocks in Figure 1)? Response: Each B block (Bottleneck layer) consists of a single linear projection layer. We have updated Section 2 to clarify the model structure.
>
> 2. Motivation and Ablation for Independent Adapters and Bottlenecks Concern: Lack of ablation studies comparing individual vs. shared adapters and bottleneck layers. Response: We have added new ablation experiments in Section 3.3.2.  The results confirm that independent adapters and bottleneck layers effectively improve performance in both concept detection and disease classification by maintaining the purity of each concept’s representation.
>
> 3. Manuscript Organization and Redundancy Concern: Redundancy between Sections 2.4.1 and 2.4.2. Response: We appreciate this structural feedback. In the revised manuscript, we have merged these sections into a single Section 2.3.2. We have also moved the entire subsection to fall logically under Section 2.3 (Concept Complement Bottleneck Model). This eliminates overlap and provides a more concise explanation of the joint learning process.
>
> 4. Response to Detailed Technical and Formatting Comments (1–13): We have revised the manuscript for all 13 points raised in your detailed comments to ensure technical and formal rigor. 1) Terminology: Corrected "CBIVLM" to CBVLM throughout the manuscript. 2) Formatting: Removed full stops from all section titles. 3) Figure 1 Consistency: Replaced the "fire" emoji with the standard "snowflake" icon for frozen modules and unified all attention abbreviations to MA (Multi-head Attention). 4) Define bottleneck layers: Provided the B-block details. 5,6）Organization: Revised Section 2.3 and 2.4. 7,8) Typos: Corrected $G$ to $g$ in Equation 6 and $\mathcal{L}_{ce}$ in Section 2. 9) Clarification: Specifically compared models' settings. 10,11,12,13) We have modified these content in the manuscript.
>
> 5. LLM Prompt Accuracy Concern: The prompt in Figure 1 appeared grammatically incorrect. Response: The prompt shown in Figure 1 was a simplified illustration. We have now provided the full, exact prompt in the Supplementary Material to ensure reproducibility.
>
> 6. Comparisons with PCBM and CBVLM Concern: Why are Concept Detection metrics and BrEaST dataset results missing for these baselines? Response: 1) Metrics: While PCBM and CBVLM provide backbone code, they lack a standardized module for concept detection evaluation. We omitted these metrics to avoid unfair comparisons resulting from manual architectural modifications. 2) BrEaST Dataset: CBVLM relies on dermoscopic concept coefficients (MEL Coefs) specifically trained for skin lesion tasks. This architecture is not transferable to breast ultrasound data, making a comparison on the BrEaST dataset invalid. 3) Configurations: We have now added the specific settings for these baselines (e.g., ResNet-50 encoder, zero-shot inference) to the Experimental Setup.
>
> 7. Intervention Thresholds (Figure 2) Concern: What are the values of the thresholds used? Response: In the revision, we implemented a fixed rule-based thresholding method for better reproducibility. We divide the range between the maximum and minimum concept logits into 10 equal intervals ($t_1$ to $t_{10}$). Metrics are calculated by progressively zeroing logits based on these thresholds across five-fold cross-validation. Detailed results and the threshold methodology are now included in the revised manuscript.
>
> 8. Statistical Information (Figure 3) Concern: Clarification on "statistical information" and reported percentages. Response: This was a typo. We have corrected the description in Section 3.

---

### Official Review · Reviewer_ad13 · 2026-01-13

**Confidence:** 4
**Preliminary Rating:** 3
**Final Rating:** 3

**Summary:**

The authors propose CCBM, a Concept Complement Bottleneck Model which leverages local concepts from medical literature to enrich the global predefined concept set. The resulting interpretable model achieves high performance in both disease diagnosis and concept detection on both dermoscopic and breast imaging datasets.

**Strengths:**

- The paper is well written and motivated
- The proposed framework introduces a clinically aligned concept bank derived from medical textbooks to ensure relevance and validity
- CCBM use concept processing with channel-wise attention, allowing each concept to have its own adapter and bottleneck layer to capture specific visual evidence effectively
- CCBM use a concept complement module to dynamically merges local concepts with a global set to enhance interpretability
- Qualitative and quantitative evaluation of the explanation
- Concept adapters are used to capture the differences in visual concepts and extract crucial features
- The model is evaluated on two modalities

**Weaknesses:**

- The method heavily relies on the first step where concepts are detected, and it is unclear whether the two steps can be trained end-to-end
- The faithfulness analysis does not really reveal the concepts' importance. Simultaneously setting several concepts scores to 0 does not provide how each concept were relevant on the decision-making process
- From visual explanations concepts seems to overlap, highlighting the need of fine-grained explanations
- The authors did not mention potential limitations of their and room for improvements
- The model architecture is a bit complex and lack clear and precise explanations
- The code is not available

**Detailed Comments:**

- From sec 2.1 it is not clear how the textual encoding lead to the visual embedding
- Is it unclear what local and global concepts refer to
- It is unclear why features images from the adapters are used as queries and the concepts embeddings used as keys and values. Are there others options to choose keys, values and queries? If yes, what motivated your choices?
- How does the k values, referring to the concept adapters is chosen as well as the lambda in the loss function?
- "Different from the previous bottleneck models (Koh et al., 2020), which directly use one bottleneck layer to get all concept scores, we use an independent bottleneck layer to get the score for each concept". While this increase the model's complexity, is there an ablation study supporting this design choice?
- It is unclear how many concepts are learned as well as how the GradCAM was generated from input with textual concepts

**Justification Of Final Rating:**

The method is interesting, as it enables learning local concepts to complement existing ones. However, the ablation studies indicate that it generally yields only marginal improvements over the baselines. In addition, the code was not released despite being requested, and no justification was provided for not releasing it. Finally, the faithfulness evaluation of the explanations is unclear and not straightforward, even though interpretability is the primary focus of the work. Taken together, these issues justify maintaining my initial rating.

**Justification Of The Preliminary Rating:**

Although the authors propose an innovative, concept-based method aimed at enriching predefined textual concepts with those acquired during training, the resulting architecture is overly complex, and the lack of code hinders reproducibility. I recommend that the authors make their code publicly available, as their work relies on publicly available reference methods.

**Questions To Address In The Rebuttal:**

See Weaknesses and Detailed Comments

---

> ### Author Response · Authors · 2026-01-25
>
> We sincerely thank you for your rigorous and constructive feedback. Your comments have guided us in refining the technical clarity and evaluative depth of our manuscript. Below are our detailed responses to each of your points.
>
> 1. Clarity on Model Architecture and Training Logic:
> Concern: The end-to-end properties of the two-stage pipeline; lack of clarity in module descriptions (attention, concept definitions); and how textual encoding leads to visual embedding.
> Response: 1) We have restructured Section 2 to clarify the model's logic. While the Concept Bank is constructed offline, the subsequent training phase is strictly end-to-end. 2) Text-to-Visual Attention (Sec 2.1): We clarified that textual encoding acts as a guidance signal (Query) to attend to local image features in a shared latent space. This filters and aggregates visual information relevant to specific clinical concepts. 3) Module Structure: Adapters, projectors, and B-blocks are now explicitly defined as independent linear mapping layers.
>
> 2. Definition of Global and Local Concepts
> Concern: Confusion regarding the distinction between "Global" and "Local" concepts.
> Response: We added precise definitions in Sections 2.1 and 2.2: 1) Global Concepts ($k$): Predefined, expert-annotated concepts provided within the datasets (e.g., standard clinical signs used across all samples). 2) Local Concepts ($m$): Sample-specific concepts discovered by the CCBM from  Concept Bank (built from medical literature and filtered by LLMs). 3) The synergy between global expert knowledge and local sample-specific discovery allows for a more fine-grained diagnostic representation.
>
> 3. Motivation for Cross-Attention Configuration (Q, K, V)
> Concern: Why are image features used as Queries (Q) and concept embeddings as Keys (K) and Values (V)? What were the other options? Response: In Section 2.3.1, we explain our choice based on primary motivations: 1) Clinical Diagnostic Logic: This setup mimics a clinician’s "image-to-text" retrieval process. A doctor observes visual patterns (Query) and retrieves corresponding terms from their knowledge base (Key/Value). 2) Semantic Filtering: By using text as K/V, the image Query performs precise semantic filtering. High attention weights only occur when a local image region aligns with a specific medical descriptor. 3) Other Options: We considered Self-Attention on concatenated features, but found it leads to excessive modality blending, compromising the semantic independence required by a Concept Bottleneck Model (CBM). Setting Text as Query often introduces background noise, as the model "searches" the whole image for a word, whereas our approach "questions" what a specific image patch represents.
>
> 4. Selection of Hyperparameters
> Concern: How were the number of adapters ($k$) and the loss weight ($\lambda$) determined?
> Response: 1) Concept Adapters ($k$): The number of global adapters matches the number of predefined concepts in the dataset. To maximize interpretability and utilize expert knowledge, we employ all available $k$ concepts. 2) Loss Weight ($\lambda$): This balances concept learning and classification accuracy. We determined $\lambda$ via a grid search from [0.1,0.2,0.5,1,10]. We have updated the manuscript details.
>
> 5. Independent Bottleneck Layers & Ablation Studies
> Concern: Does the increased complexity of independent bottleneck layers provide a performance benefit?
> Response: We added an ablation study in Section 3.3.2. The results demonstrate that independent adapters and bottleneck layers prevent feature interference between distinct concepts, significantly improving both concept detection accuracy and final disease classification performance.
>
> 6. Grad-CAM Implementation and Concept Counts
> Concern: How many concepts are learned, and how is Grad-CAM generated from textual concepts?
> Response: 1) Concept Counts: The model learns a total of $k$ (global) + $m$ (local candidate) concepts. While all $m$ candidates participate in training, for interpretability, we focus on the top-5 concepts with the highest weights for each sample (detailed in Appendix C). 2) Grad-CAM Mechanism (Sec 3.4.1): Our Grad-CAM is derived from Concept Scores, not the final class label. We use the independent bottleneck layers as the target layers. We compute the gradient of a specific concept score with respect to these feature maps. As demonstrated in our new Breast Ultrasound Case Study, different textual concepts generate distinct, specialized heatmaps for the same input image.
>
> 7. Code Availability
> Concern: Code availability.
> Response: We have provided our code link in the revised manuscript.

---

> > ### Comment · Reviewer_ad13 · 2026-02-02
> > **Implementation and explainability**
> >
> > Thank you for your answer
> > - **Model implementation**: the link to the code is not working.
> > - **Faithfulness evaluation**:
> >    - the proposed faithfulness evaluation appears unnecessarily complex. A common approach to measuring faithfulness is to mask salient regions or concepts and measure the resulting drop in model confidence. It is unclear why the authors did not adopt this as the main standard practice and instead relied on a threshold-based strategy, which appears more complicated and less intuitive. In addition, standard faithfulness analysis seems to show marginal improvement (App.C)
> >    - ablation studies (Tab3) only show marginal improve both disease diagnosis and concept detection, sometimes with higher std, meaning that the propose method does not significantly improve explanation

---

> > > ### Author Response · Authors · 2026-02-02
> > > **Official Comment by Authors**
> > >
> > > 1. Sorry for the confusion. The code has been uploaded to the GitHub repository mentioned in the text and is publicly available. We will continue to add more instructions to help readers understand and reproduce our work.
> > > 2. The setting of our inference-time intervention experiments refers to the previous work MICA [1]. We will add this citation in the revised version. Because our framework is end-to-end, the experiments of single sample intervention is not suitable.  As shown in the results of specific concept intervention in Appendix C, there are some concepts that are not decisive factors in the final decision and the intervention of these concepts may have a relatively slight impact on model performance.
> > >
> > >
> > > 3. The gain from CCBM is dependent on annotation density. On datasets with fewer concept labels (e.g., BrEaST and Derm7pt), the discovery of local concepts provides a more noticeable boost compared to the already well-annotated Skincon dataset. The primary strength of CCBM lies in maintaining high classification AUC while significantly improving the concept detection quality, which is vital for the model's interpretability.
> > >
> > > [1] Bie, Y., Luo, L., & Chen, H. (2024). MICA: Towards Explainable Skin Lesion Diagnosis via Multi-Level Image-Concept Alignment. Proceedings of the AAAI Conference on Artificial Intelligence, 38(2), 837-845. https://doi.org/10.1609/aaai.v38i2.27842

---

> > > > ### Comment · Reviewer_ad13 · 2026-02-02
> > > > **Implementation**
> > > >
> > > > The provided link (https://github.com/HongmeiWANG-HKUST/Concept-Complement-Bottleneck-Model) is not working:

---

> > > > > ### Author Response · Authors · 2026-02-02
> > > > > **Official Comment by Authors**
> > > > >
> > > > > Sorry for the mistake. The code is open access now. Thanks for your prompt response.

---

### Author Rebuttal · Authors · 2026-01-25

**Rebuttal:**

Authors' Response to Reviewer Comments
Dear Reviewers, thank you for your constructive feedback. We have significantly revised the manuscript to address concerns regarding model architecture, training logic, and fidelity.

Core Improvements: We restructured the Methods section for better logical flow, expanded the Ablation and Intervention Studies, and added visualization samples. We also provided exhaustive experimental details (concept lists, LLM prompts, baseline parameters) and an open-source link to our code.

A. Model Construction & Training Logic (Reviewer ad13, Djny, NZKk)
Concerns: End-to-end properties (ad13); unclear structure/attention/Grad-CAM details (ad13); adapter/bottleneck layers (ad13, DJny); LLM prompts (DJny); missing results/parameters (DJny, NZKk).

Revision: Section 2 now specifies that training is end-to-end following offline Concept Bank construction. Adapters and B-blocks are defined as independent linear layers. Hyperparameters and baseline settings are now detailed in the Experimental Setup. Prompts were provided.

B. Independent Bottleneck Layers (Reviewer ad13, DJny)
Concerns: Lack of evidence for independent adapters/bottleneck layers (ad13, DJny).

Revision: Added an ablation study in Section 3.3.2. Results prove independent structures prevent semantic interference and improve performance over shared architectures.

C. Intervention, Fidelity, & Case Studies (Reviewer ad13, DJny, NZKk)
Concerns: Single-concept importance (ad13); overlapping visualizations (ad13); intervention thresholds (Djny, NZKk); multi-task misalignment/fidelity (NZKk).

Revision: Added single-concept intervention tests; threshold strategy is provided; performance drops as concept scores are zeroed, confirming high Fidelity. Section 3.4.2 now includes a Breast Ultrasound Nodule Case and concept scoring for visual verification.

D. Organization & Code (Reviewer ad13, DJny, NZKk)
Concerns: Novelty vs. existing methods (NZKk); redundancy in Sec 2.4 (DJny); missing limitations (ad13); code availability (ad13).

Revision: Clarified contributions, reorganized Sections 2.3/2.4, and analyzed limitations in  Conclusion. The code link is now provided.

Additional Revision for Details:
Standardization: Corrected typos (e.g., unifying "CBVLM," "statistical information"), updated Figure 1 icons, and defined abbreviations (e.g., MA), etc.
Rigor: Re-polished the text for academic precision, ensuring qualitative terms like "significant" are only used with data support.

**Supporting Material:**

/attachment/254222f4f362ea78a67895a4d280650135f3ca34.zip

---

### Meta-Review · Area_Chair_Ao5w · 2026-02-04

**Recommendation:** Accept (Poster)
**Confidence:** 4

**Metareview:**

This paper introduces the Concept Complement Bottleneck Model (CCBM), which augments global expert concepts with local, sample-specific features mined from medical literature to enhance diagnostic interpretability. Reviewers praised the novel integration of clinically aligned concept banks and the model's robust performance across diverse medical datasets. Although initial concerns were raised regarding the necessity of independent bottleneck layers and code reproducibility, the authors successfully addressed these by releasing their code and providing ablation studies that justified their architectural choices. Furthermore, the authors refined their claims regarding statistical significance and strengthened the faithfulness evaluation in response to feedback. Given the clear methodological contribution and the improved rigor of the manuscript, I recommend accepting this paper for MIDL 2026.

---

### Decision · Program_Chairs · 2026-02-13

Accept (Poster)